

# Introducing the novel concept of cumulative concentration roses for studying the transport of ultrafine particles from an airport to adjacent residential areas.

Julius Seidler[1,3], Markus Friedrich[1,3], Christoph K. Thomas[2,3], Anke C. Nölscher[1,3]

[1]Atmospheric Chemistry, University of Bayreuth, Bayreuth, 95447, Germany
[2]Micrometeorology, University of Bayreuth, Bayreuth, 95447, Germany
[3]Bayreuth Center of Ecology and Environmental Research (BayCEER), University of Bayreuth, Bayreuth, 95447, Germany

*Correspondence to*: Julius Seidler (julius.seidler@uni-bayreuth.de)

**Abstract**

Airports are often surrounded by urban residential areas, which is both motivation and challenge for studying their potential impact on local air quality. Airports are a relevant source for ultrafine particles (UFP), which can pose a risk to human health due to their small size (particle diameter $D_\mathrm{p} \leq 100$ nm). However, in urban environments UFP originate from a multitude of biogenic and anthropogenic sources. Here, we investigate UFP in close proximity to an airport to disentangle its impact on local air quality from other urban sources.

We present observations and analysis of airborne UFP concentrations and size distributions determined at two sites in close proximity to Munich Airport. Therefore, two novel measurement stations were established north and south to the airport, but were neither situated on the axis of prevailing wind directions nor impacted by fly overs. This set-up allowed us to explore a mainly advection driven distribution of UFP into the most populated adjacent residential areas. The observation period covered a full year from August 2021 to July 2022. We analysed the dataset in three steps: (1) First, we derived UFP concentration roses using the wind data as reported in 10 m height at the airport to represent the local wind field. An increase in particle number concentrations and a shift of the modal maximum towards smaller mobility diameters became evident for wind directions including those approaching from the airport. During the airport's operation hours at daytime median particle number concentrations were 2.2 and 1.6 fold compared to nighttime at the northern and southern station. However, our data had a high variability and the direction-based analysis was uncertain due to other potential UFP sources in the surroundings and the assumption of a homogeneous, local wind field. (2) Next, we derived concentration roses employing the airflow observations from the two measuring stations at 5.3 m height. While the annual concentration rose in principle yielded the same conclusions as the first analysis step, a significant seasonal and diurnal variability of UFP and wind became evident. The influencing factors were likely other urban local UFP sources, an increased surface roughness due to green vegetation, and the atmospheric boundary layer development. (3) In order to assess the possible advection of UFP from the direction of Munich Airport relative to all other directions over the course of the year, we calculated cumulative concentration roses with both local





and site-scale wind data. Under the assumption of a homogeneous local wind field, the fraction of all UFP sampled in airflows approaching from the airport's direction was 21 % (N322) and 40% (S229). Considering a local background, the range of UFP advection from Munich Airport to the adjacent residential areas was up to 10 % in the North and 14 % in the South. It has to

be noted, that these values highlight the relative magnitude of maximum impact of the airport on local air quality, as they do not separate from other UFP sources between airport and measuring sites. Additionally, they integrate over a time period, for which the airport did not reach its full capacity compared to pre-COVID-19 times.

## 1     Introduction

Airports are a source for airborne ultrafine particles (UFP). However, their net source function and impact on the local air

quality of adjacent residential areas is not well understood. Particularly, the emission and dispersion of UFP in the atmosphere became a recent research focus, as their small size (particle diameter $D_\mathrm{p} \leq 100$ nm) enables them to enter the alveoli inside the lungs, pass on to the blood circulatory system, and may even reach the nervous system across the blood-brain barrier. Aspirated UFP are a potential risk to human health depending on their atmospheric number concentration, size, surface characteristics, and chemical composition (e.g. Ohlwein et al., 2018; Schraufnagel, 2020; Bendtsen et al., 2021).


Generally, UFP can be of biogenic and/or anthropogenic origins. Their atmospheric fate is naturally of importance to weather and climate for example by serving as cloud condensation nuclei or through their interaction with incoming solar radiation. As UFP typically dominate the atmospheric particle concentration in number, the total particle number concentration can be used as a proxy to assess their number concentrations. In the lowermost atmosphere, typical annual means of total particle number

concentrations occur between a few 1000 (Finish rural site) and several 10 000 particles per cm³ (London roadside) (e.g. von Bismarck-Osten et al., 2013; Jesus et al., 2019). Their particle size distribution exhibits a characteristic maximum at the nucleation mode or the Aitken mode, depending on the type of nearby sources. These sources can be primary as from transportation, heat and energy production, or secondary as from the nucleation of biogenic and anthropogenic gas phase precursors. Furthermore, the surrounding topography, land-use, vegetation type, atmospheric state including near surface air

flow, atmospheric stability, temperature, humidity, solar radiative forcing and oxidative capacity determine how UFP are mixed, transported and aged (Riffault et al., 2015; Oke et al., 2017; Sun et al., 2019; Trebs et al., 2023). The airborne mixture of UFP is thus highly variable in space and time, since all of these factors play a role for size, composition, morphology, and number concentration.

At airports, the sources of airborne UFP are thought to be due to ground-based processes and to aviation itself. Auxiliary power units, traffic and airplane taxiing take place on the ground (Masiol and Harrison, 2014), whereas the incomplete combustion of jet fuels or jet engine lubrication oils potentially leads to UFP release upon take-off in the air (Fushimi et al., 2019; Ungeheuer et al., 2022). As airports are typically situated in urban or sub-urban environments with a dense transportation





infrastructure, they cannot be viewed as isolated point source. Disentangling the prevalent UFP sources and transport processes with impact on adjacent residential areas thus has been the focus of airport related air quality studies worldwide. Despite heterogeneous study designs, instrumentation and airport layouts, these studies consistently showed generally increased particle number concentrations of UFP for cases of advection from the airport to the surrounding areas, for example in Los Angeles, Boston, London, Seattle, Narita, Amsterdam, Zurich, Lisbon, and Frankfurt (Westerdahl et al., 2008; Zhu et al., 2011; Hudda et al., 2014; Herndon et al., 2005; Austin et al., 2021; Fushimi et al., 2019; Keuken et al., 2015; Lammers et al., 2020; Lopes et al., 2019; Ungeheuer et al., 2021, 2022).

First evidence of airports being a potential source of UFP arose from ambient observations taken in Boston and Los Angeles. The particle number concentrations in a few hundred meter downwind of the airport were unexpectedly larger, compared to upwind, with a high variability in diameter and compositions (Herndon et al., 2005; Westerdahl et al., 2008). In the case of Los Angeles International Airport, particles with diameters of less than 600 nm had mean background concentrations of 2500 cm$^{-3}$ on the seaside of the airport (Westerdahl et al., 2008). About 500 m downwind of the airport inland the number concentration rose to a mean of 50 000 cm$^{-3}$. The modal maximum was observed for particles of the mobility diameter from 10 to 15 nm. Further studies examined the association between departing airplanes and airborne UFP number concentrations in close proximity to departure runways (e.g. Hsu et al., 2013; Zhu et al., 2011). Overall, with a regional chemical transport model study it was calculated that aviation contributed to the ultrafine particulate matter to about 7 % in downtown Los Angeles next to other regional sources such as the consumption of natural gas, on-road-traffic, and cooking (Yu et al., 2019).

Combining ground-based observations and modelling, particle mass and number concentrations were calculated for the surroundings of Zurich Airport using a Lagrangian dispersion model and mesoscale weather conditions (Fleuti et al., 2017; Zhang et al., 2020). The annual particulate mass concentrations increased only by 1 % due to aviation particles in most studied nearby locations. The background levels of particle number concentrations were increased by a factor about 2 to 10 with annual mean number concentrations ranging from 10 000 to 100 000 cm$^{-3}$ in nearby communities within 2 km distance to the airport, and up to 1000 cm$^{-3}$ in communities that were located in more than 4 km distance to the airport.

Increased particle number concentrations were also observed in greater distance, for example 40 km to the airport Schiphol Amsterdam (Keuken et al., 2015). The mean annual particle number concentration was 9600 cm$^{-3}$, showing an increase by about a factor of 3 when the wind was from the airport. Similarly, for the nearer surroundings (0 km to 10 km) of Frankfurt Airport an advection of UFP was described. Generally the particle number concentration rose, for particles with diameters of 10 nm $\leq D_\mathrm{p} \leq$ 500 nm, and the size distribution showed a maximum for particles with diameters of $D_\mathrm{p} \leq$ 30 nm, when the wind arrived from the direction of the airport during its operating hours (Ditas et al., 2022). In Raunheim, a residential area with about 5 km distance to the airport, the total and airport impacted mean number concentrations was 8600 cm$^{-3}$ and 15 090 cm$^{-3}$, respectively. These numbers were obtained before the COVID-19 pandemic times and compared to periods of reduced



airport activities during the COVID-19 pandemic. The mean number concentrations were 7990 cm$^{-3}$ in total and 11 040 cm$^{-3}$ when the wind direction was the direction of the airport. Thus, at the location, the reduced airport operations led to a decline of up to 30 % in recorded mean particle number concentrations.

Very recently close to London Gatwick Airport, size-resolved sub-micron particle observations were combined with positive matrix factorization to explore the factors contributing to particle number concentrations measured in the air at two stations with potential impact of both ground-based and aviation traffic. Here, ambient particles were characterized with mean campaign concentrations of 7500 to 12 000 cm$^{-3}$ and a peaking mode at mobility diameters about 18 to 20 nm. The airport contributed to the measured number concentration with a calculated fraction of about 17 %. However, at the two locations, more than 50 % of the detected particles originated from other traffic sources (Tremper et al., 2022).

Furthermore, a spike in the mean diurnal cycles of airborne UFP number concentrations with the start of airport activities at Luxemburg Findel Airport was observed (Trebs et al., 2023). Here, a substantial fraction was attributed to Nucleation mode particles, $1 \text{ nm} \leq D_{\text{p}} \leq 30 \text{ nm}$, hinting towards an airport contribution rather than the on-road rush-hour traffic. However, despite frequent flight activities during the day, mean UFP concentrations declined until the afternoon and increased again during the night. This diurnal variability was explained by characteristic turbulent mixing within the daytime atmospheric boundary layer and the formation of a shallow stable nocturnal boundary layer, which are either diluting or concentrating any UFP emitted into the atmosphere.

This list of studies is not exhaustive, but provides examples, which assessed the impact of airport released UFP on local air quality in adjacent residential areas. Despite a great heterogeneity, it is commonly suggested that the distribution of UFP in the local environment is particularly driven by advection and the airports' state of operation (e.g. time of day, number of flights per time, prevailing direction of departure). However, the underlying processes are generally masked by the local concert of UFP sources and atmospheric effects on their transport such as boundary layer development or the near surface wind field.

Here, we present a study that addresses the aforementioned uncertainties via three steps analysing the spatial and temporal variation of UFP and wind data for two residential areas adjacent to Munich Airport for the time period of one year. Unlike most previous studies, we conducted simultaneous observations of airborne particle size distributions in two residential areas, which were in close proximity to the airport but neither down- or upwind for the prevalent wind directions nor within fly over areas (Fig. 1). Hence, we expect the sole contribution of the airport to the local air quality in terms of UFP via advection from sources related to ground-based operations and take-offs or landings of airplanes. This novel set-up allowed us to assess

1. the overall potential impact of the airport in air quality in adjacent residential areas: testing the abovementioned view of linear advection via the annual statistics of UFP concentrations and particle size distribution with respect to wind measured at the airport in about 10 m height above ground level representing the local wind distribution,



2. the site specific spatio-temporal characteristics of this impact: analysing the seasonal and diurnal variation of UFP concentrations with respect to wind measured at the measurement site in 5.3 m height above ground level, and

3. the upper and lower limits of potential UFP advection from Munich Airport into the adjacent residential areas: comparing local and site-specific wind data with cumulative UFP for the time period of one year.

## 2    Methodology

With the aim to study the potential effect of UFP originating from Munich Airport to the residential areas in its nearest proximity, measurement stations were established at two sites for atmospheric observations in the north and south of the airport. Data and analysis reported here cover the period of one year from August 2021 to July 2022.

### 2.1    Munich Airport as study site

Munich Airport is located about 14 km north-northeast of Munich. The location can be described as rural as the surroundings are mostly under agricultural use or natural protected areas. The cities in closest proximity are Freising to the north (about 48 900 inhabitants) and Hallbergmoos to the south (about 11 100 inhabitants).

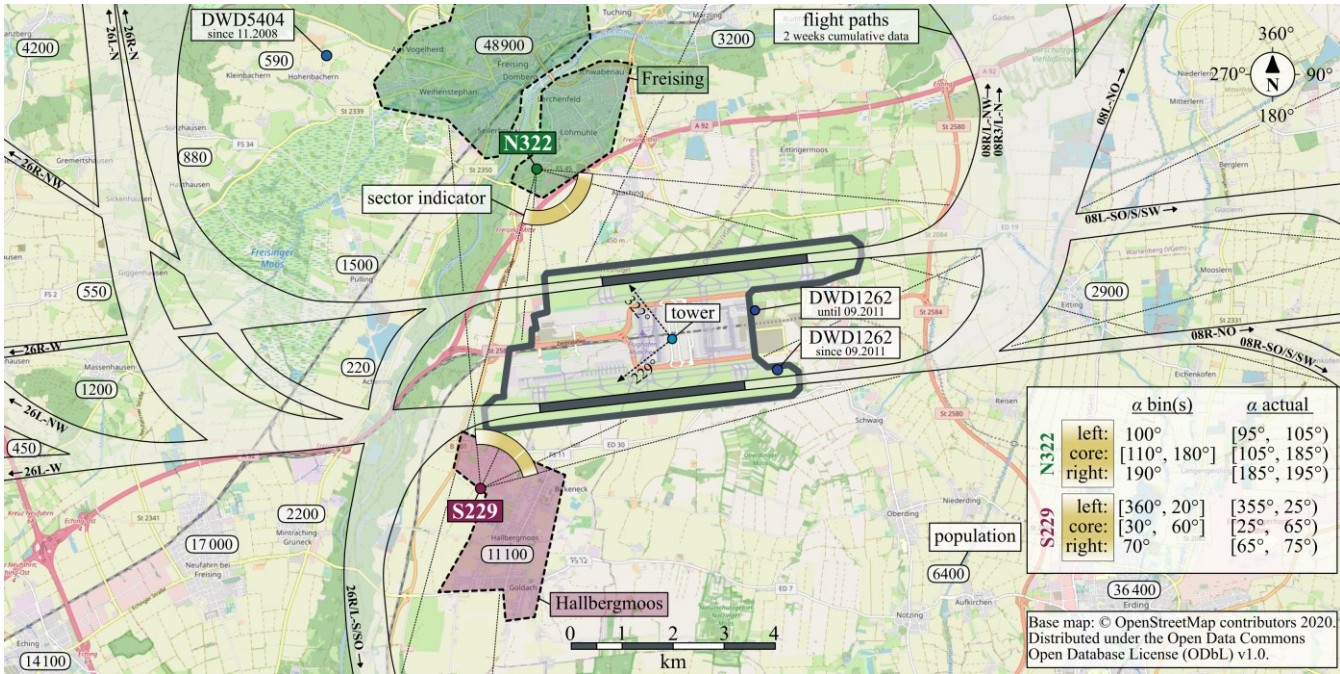

Figure 1: Map section of the premises of Munich Airport and its surroundings with the two measuring sites N322 to the north in Freising and S229 to the south in Hallbergmoos. Transparent polygons roughly mark the two populated city areas. Whitish areas mark cumulative flight paths with labels giving official corridor identifiers. Airport sector is indicated as yellowish arc. The table insert shows both sites' sector definitions with values of the 10° bins and the actual wind directions $\alpha$ that are covered by each sector part.



The airport has two parallel runways that are facing east - west, with a slight tilt of about -10°, see Fig. 1. The runway orientation is 84°/264° and falls on the axis of the two dominant wind directions. Both runways can be operated at the same time, alternating between landing and take-off. For the three years preceding the COVID-19 pandemic, the 24-h-mean was one flight per minute. Each landing or take-off is counted as one flight. We describe the geographical centre of Munich Airport by the location of its tower (48.352992° N, 11.785936° E) which is slightly shifted from the geometrical centre of the airport premises at an elevation of 450 m above mean sea level.

In the pre-COVID-19 pandemic years 2017 to 2019 Munich Airport handled from 44 to 48 Mio. Passengers per year, ranked second in Germany behind Frankfurt Airport. This ranking persisted throughout the years 2020 and 2021, that were affected by the COVID-19 pandemic (Arbeitsgemeinschaft Deutscher Verkehrsflughäfen (ADV), 2018). During these two years 11.1 Mio and 12.5 Mio passengers were handled at Munich Airport. The pandemic restrictions as well as seasonal fluctuations resulted in 17 778 flights recorded in August 2021, which is the start of the observations reported here. Cumulative flight paths were created with two weeks of pre-COVID-19 pandemic data from 2020, Fig. 1. The data was obtained from German air navigation service provider Deutsche Flugsicherung (DFS) through their web application STANLY_TRACK3.

Munich Airport is subject to night flight restrictions during nighttime from 22:00 to 05:59. This includes a ban of flights in the so-called core period, during which from 00:00 to 04:59  Based on this data, we here define daytime *with* airport activity from 05:00 to 23:59 as opposed to nighttime *without* airport activity from 00:00 to 04:59.

Table 1: Annual mean flights per hour depending on time of day for the three pre-COVID-19 years 2017 to 2019, and the two years 2021 and 2022 that include the time period covered by this study. Own calculations, data obtained from annual night flight reports by Flughafen München GmbH for the respective years. Flights per hour are rounded to full flights, except for values less than one, which are indicated as < 1.

| year | *daytime* | *nighttime* | | |
|---|---|---|---|---|
| | 06:00 to 21:59 flights per hour | *before midnight* 22:00 to 23:59 flights per hour | *core period* 00:00 to 04:59 flights per hour | *after midnight* 05:00 to 05:59 flights per hour |
| 2017 | 65 | 31 | < 1 | 8 |
| 2018 | 66 | 33 | < 1 | 9 |
| 2019 | 67 | 33 | < 1 | 8 |
| 2020 | 24 | 8 | < 1 | 4 |
| 2021 | 25 | 8 | < 1 | 3 |
| 2022 | 46 | 22 | < 1 | 5 |



## 2.2 The two measuring sites

The two measuring sites were to the north in Freising (site N322; 48.38237° N, 11.75161° E) and to the south in Hallbergmoos
(site S229; 48.32526° N, 11.73778° E). We use the airport and its tower as a geographical reference point in its surroundings.
The measuring sites are identified from the towers point of view by giving the corresponding cardinal direction and adding the
angle in degree, rounded to full degrees, see Fig. 1. Site N322 and site S229 have a shortest site-to-runway distance of 2.5 km
and 2.0 km, respectively. The direct distance between both sites is 6.4 km. Based on the flight corridors assigned by the DFS
for approaching and starting planes, none of the two stations experiences fly overs during regular operation (see cumulative
tracks in Fig. 1).

Site N322 to the north is located within the southern urban area of Freising on the premises of a city gardener and is 444 m
above mean sea level. The nearest surrounding is characterized by a business park and a highway (A92 and its feeder FS45)
spanning from south to south-east. A small meadow is located about 50 m to the west. To the north residential areas
predominate.

Site S229 to the south is placed on a wide-open space westerly of the city Hallbergmoos and is 456 m above mean sea level.
To the east and northeast residential areas can be found. When looking into airport direction north, a business park is situated
between the southern runway and site S229. The surroundings in westerly directions are mostly under agricultural use, but
crossed by a federal highway B301 in north-south direction.

## 2.3 Experimental setup

At both sites identical 10 ft laboratory containers are operated as measuring stations. These containers house the
instrumentation for determining the particle size distribution into the ultrafine range and for monitoring meteorological
conditions. The containers are climatized and set to 23 °C room temperature, the annual mean atmospheric air pressure is
965 mbar.

Meteorological data, including wind speed and direction, was measured by a compact weather sensor (Lufft, WS700-UMB).
The sensor was mounted on top of an aluminium pole, placing the sensor 2.2 m above the container roof and 5.3 m above
ground level at the container corner farthest from the $PM_{10}$ inlet (see below). The weather sensors have been mounted with a
deviation of a maximum of 10° to the north. The remaining deviation is corrected by continuous position monitoring by the
sensor's built in compass. The local declination was calculated to be 3° during the observation period covered in this study
and accounted for.



From outside the container air is probed via a PM$_{10}$ inlet (R&P, RP57-000596) designed for 16.7 L min$^{-1}$. The PM$_{10}$ inlet
allows undisturbed sampling of air with particles of aerodynamic diameters less than 10 μm at a height of 1.1 m above the
container roof and 4.2 m above ground level. The inlet is maintained at an overall flow rate of 16.7 L min$^{-1}$ and connected to
an isokinetic flow splitter (Dockweiler, MB6709) inside the container by seamless stainless steel tubing (1.25 in × 0.065 in).
From the flow splitter a sub stream of air passes a 3/8 in membrane dryer (TROPOS, 300 mm Nafion dryer) and is then
connected to the inlet of a mobility particle size spectrometer (MPSS) by conductive silicone tubing (TSI, 3001788). The
overall distance passed by probed air from PM$_{10}$ inlet to MPSS inlet is about 3.1 m and results in a residence time of about 7 s.

Particle size distributions were measured with the MPSS (TROPOS). Each MPSS was equipped with a modified Hauke
medium type differential mobility analyser (DMA) and a 370 MBq neutralizer with $^{85}$Kr (Eckert & Ziegler Cesio, NER 8275).
In each MPSS a CEN-certified condensation particle counter (CPC) with a diameter for 50 % counting efficiency $D_{p,50} = 7$ nm
was used (TSI, 3750). A full scan over the mobility diameter range of 10 to 800 nm took 5 min and combined one up- and one
down-scan, each with a resolution of 71 bins, 43 of which cover the mobility diameters from 10 to 100 nm. The nominal
aerosol flow rate through the MPSS is 1 L min$^{-1}$. The aerosol to sheath air flow rate ratio is 1:5. All membrane dryers are
operated in countercurrent flow using particle free and dry pressurised air.

For operation and maintenance of the MPSS we followed standard protocols as for example established by Wiedensohler et al.
(2012). Every 3 to 4 weeks the MPSS was calibrated with a Latex-Standard (203 (4) nm, Thermo Scientific, 3200A) to check
sizing accuracy/sheath air flow and plumbing time, high voltage output and flows were evaluated and re-adjusted if necessary.
Checking zero air and flow was performed for both MPSS and CPC separately.

### 2.4   Wind data

Meteorological conditions at the airport are continuously monitored by the German Weather Service (Deutscher Wetterdienst,
DWD) since 1992, with a minor relocation of the monitoring station DWD1262 on the airport premises by about 1 km in
September 2011 to its current location (445 m above mean sea level; 48.34771° N, 11.81338° E). The 10-years mean from
2010 to 2019 (see Fig. SI1) has two maxima in directions west and east highlighting the two dominant wind directions, which
occur in 41 % and 26 % of the time, respectively. Calm conditions with wind speed < 0.3 m s$^{-1}$ were observed in 2.2 % of all
times. The wind rose within the herein reported observation period was generally similar to the 10-years mean with west and
east winds occurring in 43 % and 22 % of the time, and calm condition in 1.1 % of all times (Fig. 2 and SI1). The DWD is
operating another monitoring station DWD5404 in Weihenstephan-Dürnast about 10.7 km northwest of DWD1262 (477 m
above mean sea level; 48.40253° N, 11.69457° E). Despite their differences in distance and altitude, the west-east-pattern
observed for DWD1262 is also found at DWD5404. The DWD reports wind data based on a wind rose with 35 bins ranging
from 10 to 360°. The actual bin width is -5.0°/+4.9°. Calm conditions are set to 0°.



Here, we use the airport's wind data (DWD1262) as reference and representation for the local wind field. As we have wind data from our measuring sites as well, we can explore the effect of the site-specific, micro-scale wind field for comparison. At each site the station's wind data is retrieved in close proximity to the particle sampling in 5.3 m height. Due to a shifted start

of the meteorological observations, this dataset is only complete for November 2021 to July 2022. We handled our data according to the standards of DWD using the same 35 bins and reporting calm conditions in the center of each wind rose, see Fig. 2. The meteorological data measured at the measuring sites N322 and S229 is processed according to DWD quality guidelines (Lanzinger et al., 2021).

Based on the wind direction binning, we defined a sector for each site that is considered to be under potential airport influence when assuming linear advection. The sector is made of a core covering the projected width of the runways for each site, but without leaving the airport premises. The sector parts left and right include the wind direction bins needed to cover the remaining area of the airport premises completely. A schematic representation and the resulting sectors are included in Fig. 1.





Figure 2: Wind roses based on local scale wind data from DWD1262, source DWD, (left column) and site-scale wind data from sites N322 (middle column) and S229 (right column) for the period of this study (top row) and three seasons covered by it (Winter 2021/2022, Spring 2022, Summer 2022). Bottom right corner gives the maximum wind speed observed for each data set. Autumn is not shown due to missing site-scale data, see section 2.4.




## 2.5 Data processing and analysis

Post-processing the MPSS data: We followed standard protocols as for example established by Wiedensohler et al. (2012) with inversion from particle mobility to particle number size distributions via bin width normalization and multiple charge

correction. The data was corrected for the particle losses considering the equivalent lengths and individual flow rates of all flow-through components from $PM_{10}$ inlet to the CPC inside the MPSS. Further the data was corrected for internal losses due to diffusion and the individual CPC counting efficiency. In accordance with ACTRIS calibration standards from intercomparison workshops in December 2020 and March 2023 the overall uncertainty is ±10 %.

Particle number concentrations, size distributions and modal particle diameter: All particle number concentrations in this study are reported non-normalized as $N$, not as normalized $\Delta N/\Delta \log D_{\mathrm{p}}$. The unit is particles per centimetre cubed, $[N] = cm^{-3}$. The MPSS recorded particle size distributions, which can be used to assess various characteristics of the ambient aerosol, see Fig. 3. Firstly, we determined the number concentration of all particles between the mobility diameters 10 to 800 nm, being $N_{800}$. The UFP number concentration was derived from the interval of mobility diameters between 10 to 100 nm, being $N_{100}$. Secondly,

the MPSS particle size distribution was used to derive the modal particle diameter.

Expressing uncertainty: If not stated otherwise, we express the uncertainty of mean of measurements as standard deviation noted as *mean (standard deviation)*. The relative standard deviation/the coefficient of variation $c_{\mathrm{v}}$ is the ratio of standard deviation to mean and is reported as $[c_{\mathrm{v}}] = %$.

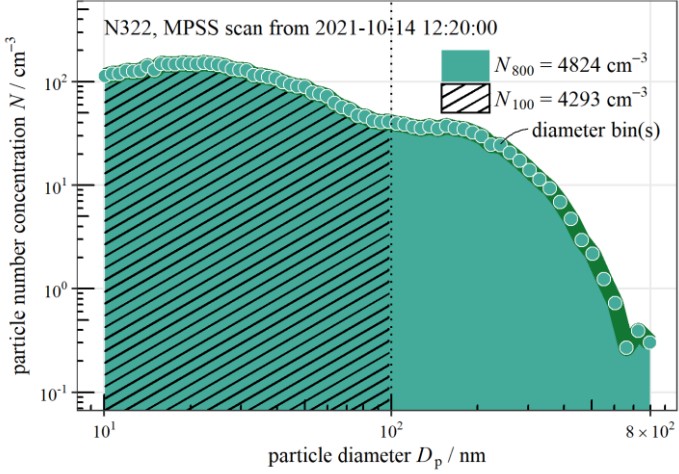

Figure 3: Single MPSS scan from site N322 from the period of this study over the particle mobility diameter range of 10 to 800 nm. Diameter bins are marked at their midpoints. The colored and hatched areas represent the integrated ranges for calculating $N_{800}$ and $N_{100}$ from $N$.




Date and time: All dates and times are reported as local time, which means either CET/UTC+01 or CEST/UTC+02 depending on the actual date-time pair.

Annual and seasonal statistics: These are reported here as means with the standard deviation, as median values and the
interquartile range, i.e. the distance between the 25 % and 75 %-percentiles. The statistical distribution of the particle number concentrations is non-Gaussian, which can be seen from Fig. 4. Thus, the mean is increased by the extremes in the analysed data set and we present the median as well. The median is an adequate measure to describe the statistical distribution and the typical particle concentrations around the year or the seasons. For the latter, the meteorological definition of the season was used with the start of spring on 01. March, start of summer on 01. June, start of autumn on 01. September, and start of winter
on 01. December. Due to the missing meteorological data in autumn, only winter, spring and summer have been analysed in detail for the site-scale view. Note that when reporting values for summer 2022, this covers only the first two months, June and July, but not August, see also Fig. 10 and Fig. SI3 - 5.

Concentration rose: For deriving the relation between the observed particle number concentrations and the wind directions,
we combined these data-sets as concentration rose by polar box plots or so called "squeeze box plots". Throughout this study, we present two versions of the concentration rose, as we aim to evaluate the commonly applied strategy of using the wind data monitored at the airport (local scale) and explore the effect of using the measurement stations' wind data at each site (site-scale). For the local scale, first the wind as monitored at the airport is combined with the observed particle number concentrations. The wind is measured at 10 m height and with no distortions of flow. Because the topography around the

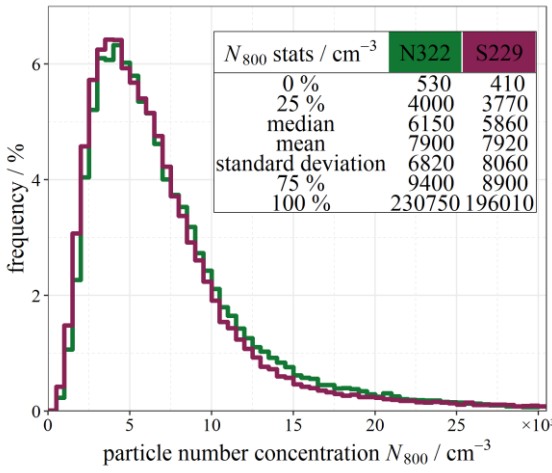

Figure 4: Frequency histogram and annual statistics of $N_{800}$ particle number concentrations at the sites N322 in Freising and S229 in Hallbergmoos for the observation period of this study. The plot is cut at $N_{800} = 30\,000$ cm$^{-3}$. Bin width is 500 cm$^{-3}$. The table contains descriptive statistics, rounded to full multiples of 10.



airport is a plane, it can be assumed that the wind field is rather homogeneous at that height. To test the effect on sampling within the increment of the supposedly urban background environments of Freising and Hallbergmoos, we assessed the site-scale wind data of our measurement sites to derive concentration roses.

Cumulative number concentrations: The presented data covers the annual cycle of the observation period from August 2021
to July 2022 almost completely. Missing data only occurred due to scheduled maintenance and 12 days of an intensive measurement campaign 25. April to 06. May 2022. This yields a data availability of 92 % for site N322 and 94 % for site S229. This data-set allows us to sum up the observed particle number concentrations to a cumulative value. The observed period was special in a way, that during the COVID-19 restrictions no fireworks were allowed in Germany for celebrating New Year's Eve. Therefore, no extreme particle emissions occurred during that night and the dataset is a good representation of typical
airborne particle loads at the two sites during the course of one year. From the cumulative data a relation to the wind direction frequency can be derived as cumulative concentration rose as follows: First, we normalised the cumulative particle number concentration to the overall sum of all particles observed at the measurement locations after the duration of one year. The overall sum of observed particles after one year thus corresponds to 100 %. Next, we combined the data of both measurement sites with the wind direction as monitored at the airport or the measuring sites.


Except for the map in Fig. 1 all geographical data (positions, distances, angles, and elevations) was obtained by combining base maps from OpenStreetMap (© OpenStreetMap contributors, www.openstreetmap.org/copyright) with base maps and orthophotos from the Bavarian Surveying and Mapping Authority (www.geodaten.bayern.de, CC BY 4.0, www.creativecommons.org/licenses/by/4.0/) in GIS-Software.

## 3    Results and discussion

During August 2021 to July 2022, the $N_{800}$ particle number concentrations varied from 410 to more than 190 000 cm$^{-3}$, with a tendency to smaller number concentrations in the winter at both sites, Fig. 5 (a) and (b). We observed significant differences in atmospheric particle number concentrations between the two stations on an annual basis. The distribution was non-Gaussian from Anderson-Darling-test with $p_{N322} \ll 0.05$ and $p_{S229} \ll 0.05$, and both sites featured non-similar distributions from
Kolmogorov-Smirnov-test with $p \ll 0.05$, see Fig. 4. The annual mean $N_{800}$ particle number concentrations were similar for both sites with 7900 (6820) cm$^{-3}$ and 7920 (8060) cm$^{-3}$ at site N322 and S229, respectively. The annual median of $N_{800}$ particle number concentrations was 6150 cm$^{-3}$ for site N322 and 5860 cm$^{-3}$ for S229. These median particle concentrations are similar to





Figure 5: Overview of the entire data sets for August 2021 to July 2022 presenting the time-lines of particle number concentrations $N_{800}$ at sites N322/Freising (a) and S229/Hallbergmoos (b) with running boxplots per month, the sum of flights per month at Munich Airport (c), the site-scale wind measured at sites N322 (d) and S229 (e), and the local scale wind recorded by DWD1262 at Munich Airport, source DWD (f). Calm conditions $\alpha = 0°$ are indicated separately for (d - f). Maximum wind speed for is $ff_{max} = 21.3$ m s$^{-1}$ for (f).



other German urban background stations for which a multi-annual median of about 4300 to 7400 cm$^{-3}$ (for $D_p$ = 20 to 800 nm) was reported by Sun et al 2019. For both sites, the median fraction of $N_{100}$ in $N_{800}$ particles was 85 (9) %.

As the number of flights per month at Munich Airport seemed to exhibit a similar seasonality than the monthly mean particle number concentrations for $N_{800}$. We tested whether the data sets can be correlated but did not find a linear relationship with $R^2_{N322}$ = 0.12 and $R^2_{S229}$ = 0.02. For the median of $N_{800}$ the results were the same with $R^2_{N322}$ = 0.23 and $R^2_{S229}$ = 0.15. Generally, the relatively lower $N_{800}$ particle number concentrations in wintertime occurred within stormy periods when highest wind speeds were reached (> 20 m s$^{-1}$ for wind measured in 10 m height, DWD1262) and when wind from the west was most frequent. Figure 2 and 5 present as well the wind data determined at the airport by DWD1262 and at the two sites, N322, and

S229. The latter are relatively more affected by vegetation and building induced surface roughness when compared to the 10 m wind data determined at the airport. For example, at the site N322 in Freising, the frequency of observed wind directions displays features besides to the prevailing wind directions and maximum wind speeds were measured about a factor of 3 smaller than at the airport, see Fig. 5. Calm wind conditions were monitored more frequently, occurring for 32.3 % (N322) and 15.8 % (S229) of all recorded data points during the observation period compared to 1.1 % at the airport, see Fig. 2.

These datasets were the basis for the following combined analysis in three steps from local to site-scale view leading to a novel approach via cumulative concentration roses.

### 3.1 Integrated, local view on overall potential impact of the airport on air quality in adjacent residential areas

For exploring the variation of particle number concentrations with wind direction, a concentration rose based on the wind data

as monitored at the airport itself (local scale) and the particle number concentrations as observed at sites N322 and S229 was calculated and displayed in Fig. 6. Here, the wind measured at Munich Airport in 10 m height above ground was used to represent the local wind distribution. The combination with the sites' $N_{800}$ particle concentration shows an increase for the observations at site N322 for wind directions between 95 to 225°. At site S229 in Hallbergmoos, the increase is of similar order but the wind directions are 305 to 55°. For both sites, the wind directions with generally increased particle number

concentrations include those that we defined as winds with possible impact by Munich Airport, see Sect. 2.1. Assuming linear advection and a homogeneous local wind field, our sites could have received wind from Munich Airport. If we compare the wind directions attributed with influence by Munich Airport with all other wind directions within the observation period of one year, the median $N_{800}$ particle number concentrations about a factor of 1.7 and 1.5 higher in Freising and Hallbergmoos.



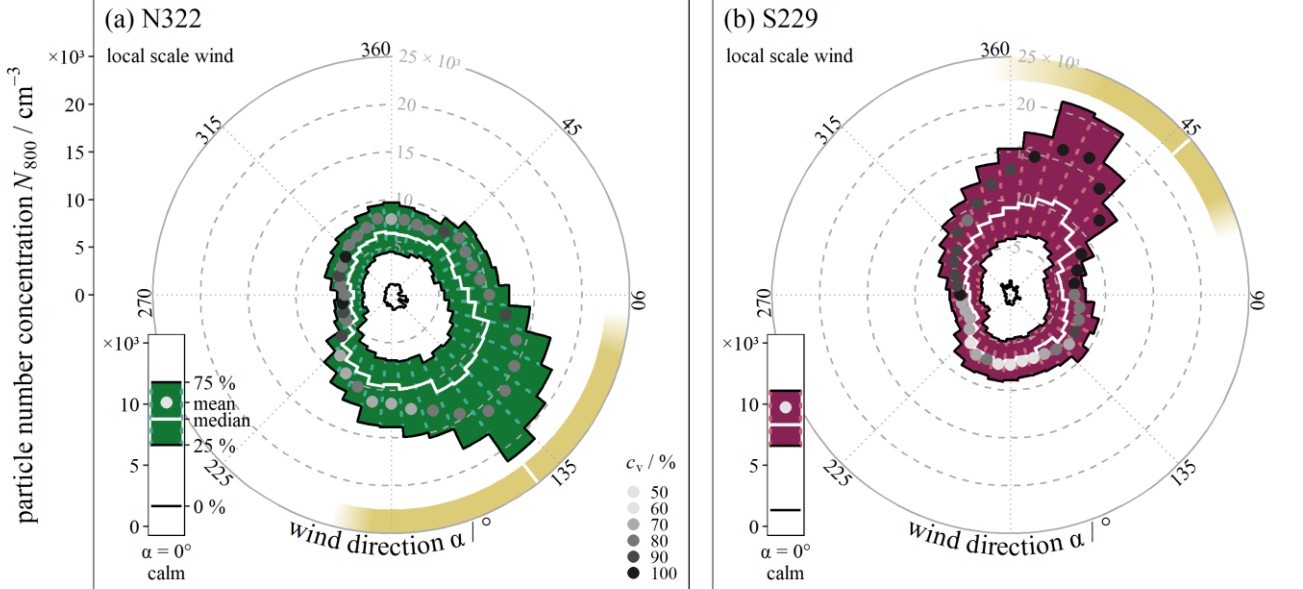

Figure 6: Concentration roses as squeeze box plot of local scale wind data measured at the airport by DWD1262, source DWD, and statistics for $N_{800}$ number concentrations for sites N322 (a) and S229 (b) for the observation period of this study. Boxes represent the 0 percentile, 25 %-percentile, median, mean (position) combined with relative standard deviation $c_v$ (grey shades coded), and 75 %-percentile. The inset plot pictures the same statistics for calm conditions. Angular bin width is 10°, the bins are centered. The yellowish arc is the airport sector indicator, see Fig. 1.

The same tendency was found for airborne UFP ($N_{100}$) number concentrations with respect to local wind. Figure 7 displays concentration roses for daytime, recorded only during the airport operation hours, and nighttime, when no flights were scheduled, see Sect. 2.1. Daytime median $N_{100}$ number concentrations are about a factor of 2.2 and 1.6 higher for wind directions arriving from the airport compared to all others for sites N322 and S229, respectively. During nighttime this is reduced to 1.4 (N322) and 1.2 (S229). Interestingly, the relative standard deviation is much lower during nighttime. The relative standard deviation is 62 % as opposed to 98 % during daytime at site N322 and 70 % as opposed to 115 % during daytime at S229 considering all measured UFP number concentrations and all wind directions. Furthermore, the particle size distribution varied characteristically between day and night.

Figure 8 presents a shift for the annual median maximum in the particle size distribution towards larger mobility particle diameters during nighttime (25 to 50 nm). During daytime, the smallest modal particle diameters were 10 to 15 nm and observed at site N322. These occurred for the same wind directions that received highest annual median particle number concentrations. Interestingly, at site S229 in Hallbergmoos the modal diameters were 15 to 25 nm for all wind directions except for easterly winds. Here, the median modal particle diameters were greater than 30 nm and the fraction of $N_{100}$ in $N_{800}$ reaches





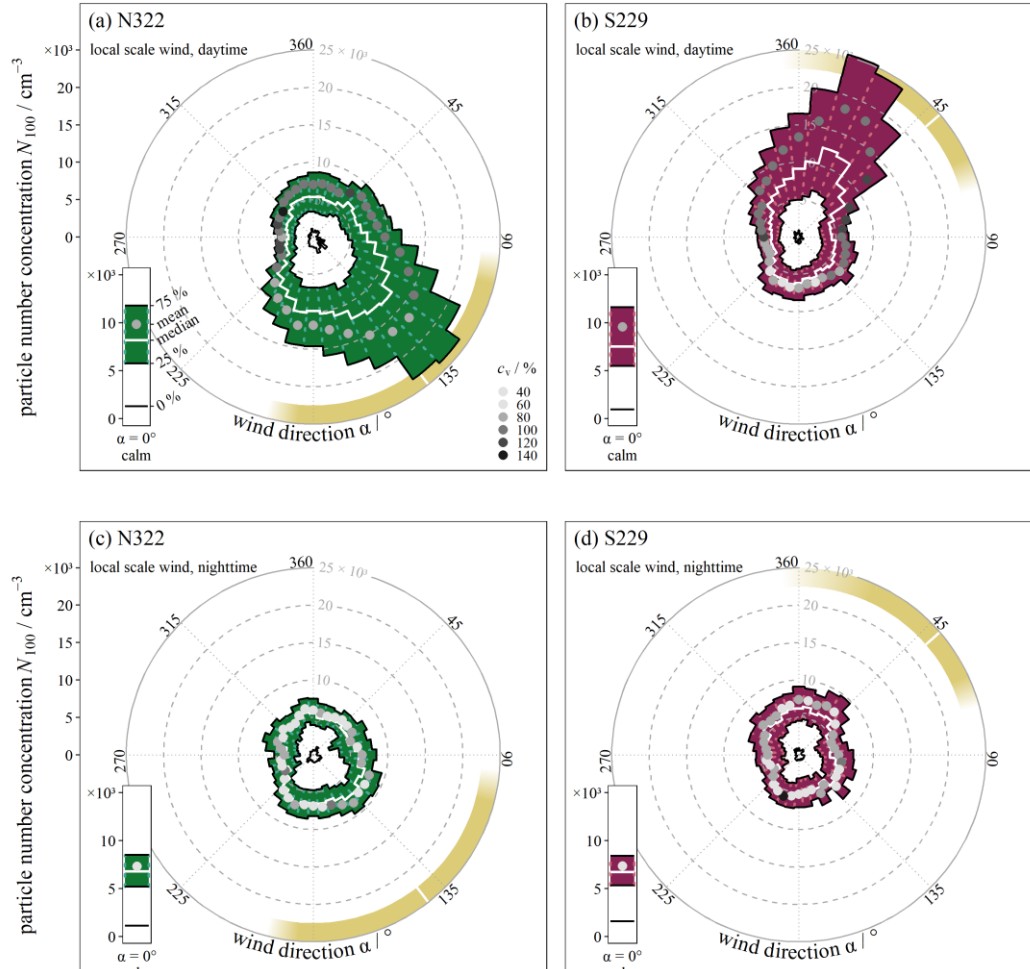

Figure 7: Concentration roses as squeeze box plots of local scale wind data as measured at Munich Airport, source DWD, and statistics for particle number concentrations $N_{100}$ for sites N322 (a and c) and S229 (b and d) for the observation period of this study. Top and bottom panel distinguish between daytime and nighttime as defined in Sect. 2.1. The yellowish arc is the airport sector indicator, see Fig. 1.

overall smallest values of about 80 % (Fig. SI2). Considering only daytime for August 2021 to July 2022, the median ratio of

$N_{100}$ to $N_{800}$ particle number concentrations was 92 % in Freising and 88 % in Hallbergmoos for observations with potential wind from the airport. For all other directions, it was 87 % and 88 % at the stations in Freising and Hallbergmoos. Within the given variability, the difference was insignificant with $p_{N322} \ll 0.05$ and $p_{S229} \ll 0.05$ from Wilcoxon rank sum test (see Fig. SI2).

After exploring the effect of the local wind field on the potential transport of UFP from the airport into the adjacent residential areas, we find three important points of discussion: (1) For airflow from the airport previous studies highlighted a shift towards



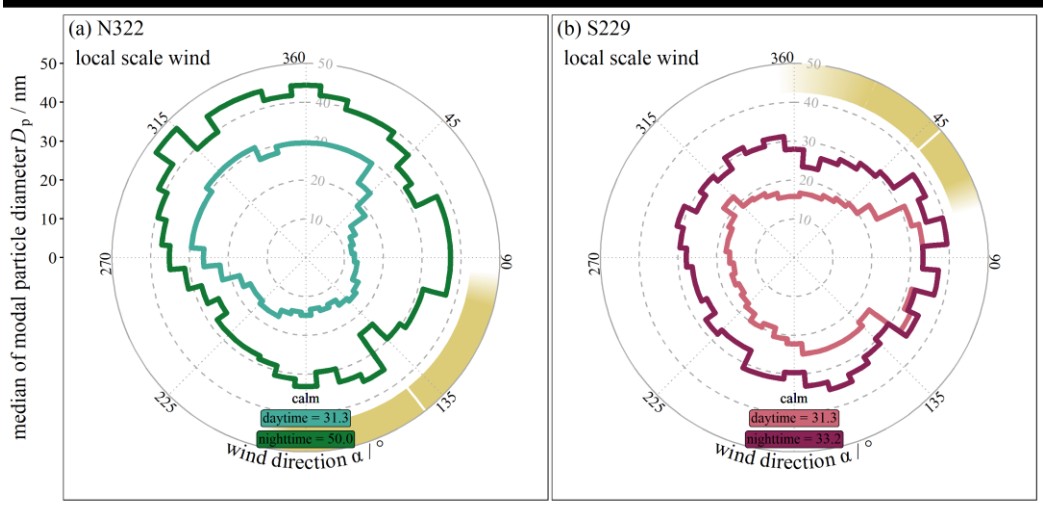

Figure 8: Median of modal particle diameter $D_p$ as derived from the particle size distribution during the entire observation period for sites N322 in Freising (a) and S229 in Hallbergmoos (b) for local scale wind data, source DWD, and depending on daytime/nighttime. The yellowish arc is the airport sector indicator, see Fig. 1.

smaller particle diameters, mainly to $D_p < 30$ nm (Keuken et al., 2015; Fushimi et al., 2019; Rose et al., 2020; Ungeheuer et al., 2021; Ditas et al., 2022). Our findings show a shift to 10 to 25 nm as well, however, we cannot precisely attribute it to the defined wind directions with potential airport impact. Particularly for the southern site in Hallbergmoos a wide range of wind

directions was associated with such small modal diameters. Possibly, this is related to airplanes taking-off during west winds to southern destinations (see Fig. 1 and corridor 26R/L-S/SO). (2) Our view on the variation of $N_{800}$ particle number concentrations with wind direction generally agrees with the findings of other studies that established the view of a mainly linear advection driven transport of particles from the airport in the surroundings (e.g. Ditas et al., 2022; Keuken et al., 2015). At the two sites studied here, the annual median particle concentrations for $N_{800}$ were about a factor of 1.7 and 1.5 (N322 and

S229, see Table SI1) higher for times with wind from the direction of the airport relative to all other wind directions. Calculating the same for the annual mean particle concentrations results into a significant increase of 1.9 to 2.0 (Welch two sample $t$-test for large non-Gaussian samples, $p_{N322}$ and $p_{S229}$ << 0.05). However, a large relative standard deviation for $N_{100}$ of $c_{v,N100} = 112$ % and $c_{v,N100} = 95$ % for sites N322 and S229 highlight a strong variability in the time series. (3) Thirdly, we observe a dependence on time of the day similar to Frankfurt Airport (Ditas et al., 2022). At daytime, likely the higher

variability and elevated particle number concentrations are resulting from the airport's operations. Yet the analysis of the concentration roses doesn't allow differentiation from urban activities in close proximity to the measuring sites such as rush-hour traffic. Approaching the variability in the observed particle number concentrations in combination with the wind data from Munich Airport assuming a local, homogeneous wind field thus serves here only as a first estimate. This is a common approach for describing the transport of particles in the atmosphere in close proximity to a source, however simplifies the

contribution of other atmospheric processes as drivers for particle transport, such as turbulent mixing and the development of



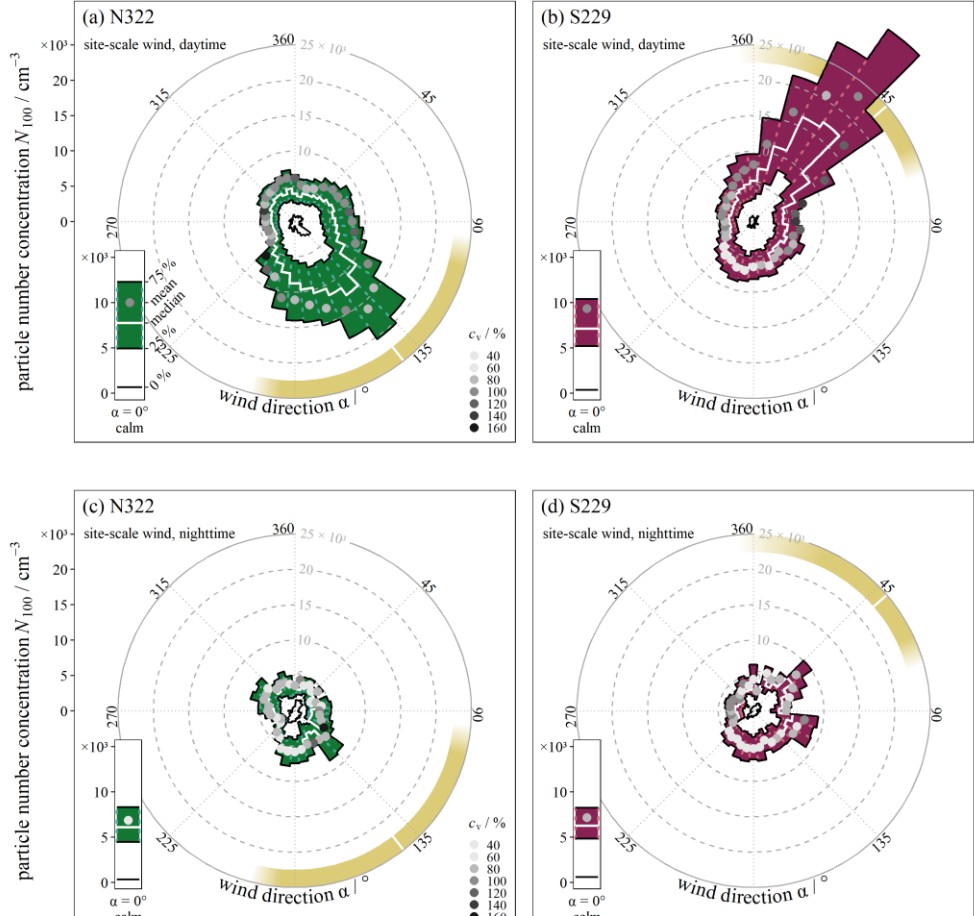

Figure 9: Concentration roses as squeeze box plots of site-scale wind data as measured directly at the sites and statistics for particle number concentrations N100 for sites N322 (a and c) and S229 (b and d) for the time period of November 2021 to July 2022. Top and bottom panel distinguish between daytime and nighttime as defined in Sect. 2.1. The yellowish arc is the airport sector indicator, see Fig. 1.

the atmospheric boundary layer. For this reason, we next explored whether we could add precision to results and discussion when considering the variability of wind as monitored at the measuring stations directly.

## 3.2 Detailed view on the site-specific spatio-temporal characteristics with respect to air quality in airport adjacent residential areas

To test whether a more detailed view on the site-scale would confine the previous findings or lead to different conclusions, we used here the measuring stations' wind data determined at 5.3 m height. Thus, Fig. 9 presents the same concentration roses as displayed in Fig. 7 for $N_{100}$ particle number concentrations, but with the wind data from the respective measuring sites. In comparison, the previously described main features persist: The concentration rose exhibits an elevation in the same directions for the entire year. In Freising, median $N_{100}$ particle number concentrations were significantly increased by a factor of 2.0



during day and 1.5 during night (Wilcoxon rank sum test with $p \ll 0.05$) when comparing wind directions with potential impact of Munich Airport to all others, except for calms (see. Table SI1). In Hallbergmoos, the factor of increase was 1.8 during day and 1.3 during night (Wilcoxon rank rum test with $p \ll 0.05$). The greatest difference is the lower variability during nighttime, which can be seen from the lower relative standard deviation at sites N322 and S229 of about 67 % and 75 %.

From Fig. 2, we further notice a characteristic seasonality of the sites' wind roses. While the wind rose as monitored in 10 m height at Munich Airport (DWD1262) falls into the two main wind directions for all defined time periods, the entire time series, winter, spring and summer, the wind rose measured at 5.3 m height at the site in Freising was different for spring and summer. The wind rose determined at the southern station in Hallbergmoos (S229) deviated in summer from that across the entire year. The predominant wind direction shifted. For example, in spring 2022 for site N322 four maxima could be noted

and for site S229 the wind approached relatively more frequently from the northeast. Generally, wind speeds were reduced at the two measuring sites compared to those at the airport. The occurrence of calm conditions was greatest in summer with 38.5 % (N322) and 25.5 % (S229). Likely, the additional surface roughness by vegetation and buildings impacted wind directions and speed, and hence the transport of airborne particles close to the ground.

This seasonality is pictured also in the concentration roses, when separating winter, spring and summer from the entire observation period (Fig. SI3 - 5). The main message persists and an increase can always be noticed. However, its amplitude varies from maximum values in winter time in Freising, with a factor of 3, for $N_{100}$ considering day and nighttime data, to minimum values of 1.3 for summertime in Hallbergmoos.

Similarly, the diurnal variation of each site's particle concentrations was prone to seasonality as presented in Fig. 10. Both sites exhibit a similar diurnal variation, despite different site-specific surroundings. $N_{100}$ particle number concentrations increase during the morning rush-hour reaching a maximum at 07:00 and 08:00 for summer and winter, respectively. In winter, a second, more pronounced peak can be noted in the evening (maximum means at 18:00 to 20:00). In summer, the second maximum was recorded later at around 21:00 to 23:00. For the example of site N322 (Freising), the wintertime maximum

median $N_{100}$ concentrations reached 5610 cm$^{-3}$ in the morning at 08:00 and 6720 cm$^{-3}$ in the evening at 19:00. We note that the summertime $N_{100}$ particle concentrations were generally elevated compared to wintertime reaching the two maxima with median values of 8260 cm$^{-3}$ (07:00) and 8500 cm$^{-3}$ (23:00). In the afternoon median $N_{100}$ particle number concentrations declined at both sites. It is interesting to look at the difference between mean and median values, as the means are driven by extremes, for example by events with high particle number concentration but short duration. The first periods with noticeable

differences between mean and median are the morning and evening rush-hour times for both winter and summertime. The second period is the summertime afternoon, when median values declined but means increased.



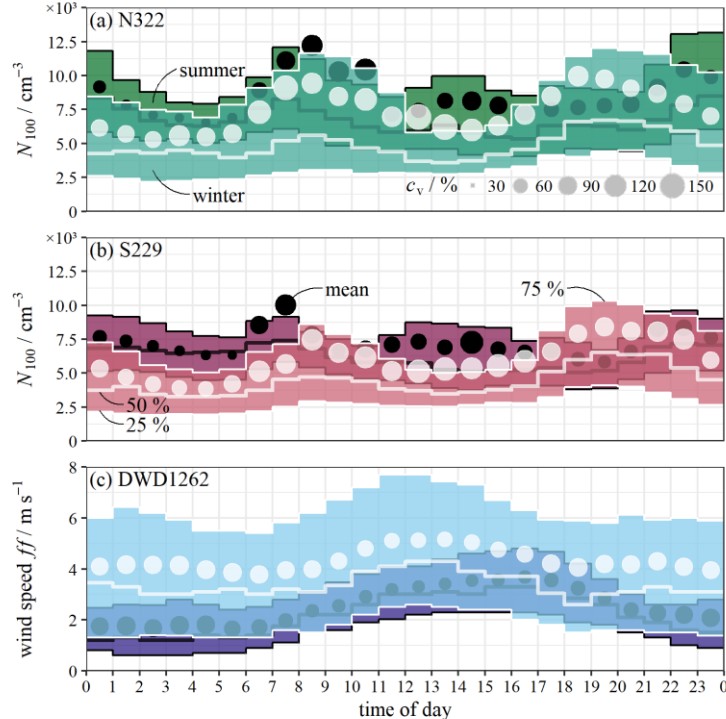

Figure 10: Diurnal variation as one hour box plots of number concentrations $N_{100}$ for sites N322 (a) and S229 (b) for local scale wind data from DWD1262, as well es scalar wind speed $ff$ measured by DWD1262, source DWD (c) for summer and winter. Maximum and minimum values are not shown. Given are data for Summer 2022 and Winter 2021/2022 from our observation period.

Based on these observations, we discuss the observed spatio-temporal variability of the particle number concentrations at the two sites: The diurnal variation of particle number concentrations likely displays the interplay of a typical urban air mixture
between local particle sources and atmospheric processes of transport and aging during to the development of the boundary layer. First, the deviation between means and medians provides evidence for local sources, which cause non-Gaussian statistics and higher skewness. Typical local sources other than the airport include road traffic, residential heating, photochemistry leading to particle formation, and agricultural activities. Previous studies attributed the two peaks in the morning and evening to particles emitted within the rush-hour traffic (e.g. von Bismarck-Osten et al., 2013; Jesus et al., 2019; Tremper et al., 2022)
Residential heating likely contributed to the observed particle number concentrations in winter (Yu et al., 2019). During summertime afternoons possibly two effects might have increased the hourly mean $N_{100}$ particle number concentrations: (1) photochemistry leading to new particle formation and/or (2) airport UFP emissions. Secondly, the development of the atmospheric boundary layer typically leads to a larger mixing volume during the daytime compared to a shallow stable layer during nighttime. Hence, during daytime emissions are diluted rapidly and mixed via turbulent and advective air movement.
During night, mixing and transport can be reduced (Hudda et al., 2014; Trebs et al., 2023). These effects can be noticed for example for summertime at site N322, when the rush-hour peak of mean $N_{100}$ particle number concentrations declined almost twice as rapidly during the day than during the night (decline rate of hourly means: morning (08:00 to 12:00) = 1260 cm$^{-3}$ h$^{-1}$,



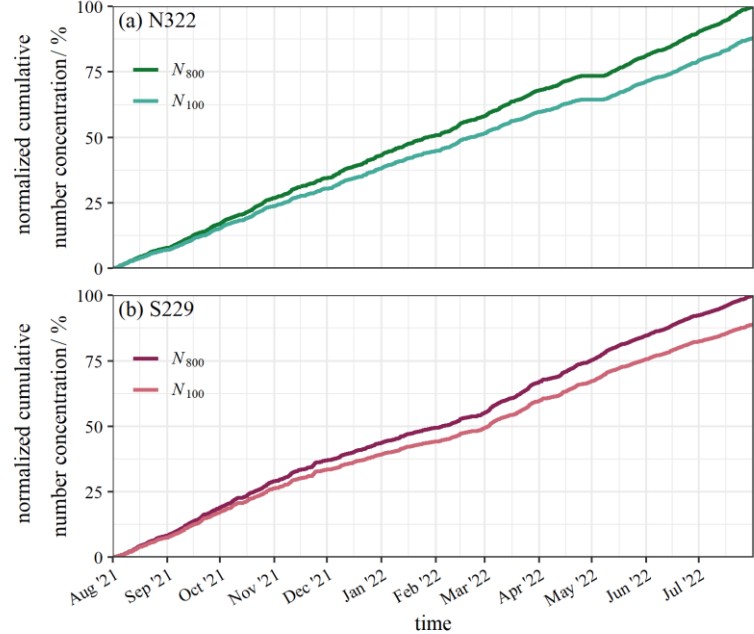

Figure 11: Normalized time series of relative cumulative number concentrations $N_{800rc}$ and $N_{100rc}$ for sites N322 in Freising (a) and S229 in Hallbergmoos (b). The data used is the same as in Fig. 5, but both $N_{100}$ and $N_{800}$ are expressed relative against the maximum cumulative total number concentration for $N_{800}$ at each site. Time resolution is 5 min.

evening (22:00 to 04:00) = 710 cm$^{-3}$ h$^{-1}$). Similarly, a decline of UFP during daytime despite a high frequency of flights was reported from Trebs et al. (2023), for the airport in Findel, Luxembourg. Furthermore, we observed larger median modal
diameters during the night for both sites up to 50 nm, see Fig. 8, which is an indication that relatively more aged, and not freshly emitted or formed particles were detected at these times.

Overall, the diurnal variation of UFP number concentrations at the two sites does not provide any strong evidence for the impact of the airport on the local air quality. In contrast, the concentration roses for winds blowing from the direction of the
airport had the highest daytime UFP number concentrations. Local and site-scale concentration roses agreed on this view when considering the entire data set over the period of one year, see Fig. 7 and Fig. 9. However, we find that the concentration roses lack the information about the potential background particle load, additional sources in the same direction and, most importantly, the frequency of occurrence of a wind direction.

### 3.3 Upper and lower limits of cumulative UFP transport from Munich Airport into adjacent residential areas
considering both wind on local and site-scale

As the presented data covers the annual cycle of the observation period August 2021 to July 2022 almost completely, it can be used to sum up the observed particle number concentrations to a cumulative value, see Fig. 11. The normalized cumulative number concentrations of $N_{800}$ and $N_{100}$ particles rise constantly throughout the year. Furthermore, the total cumulative number



concentrations are similar for both locations ($N_{800}$ = 784·$10^6$ cm$^{-3}$ and 759·$10^6$ cm$^{-3}$ at sites N322 and S229). The fraction of

cumulative $N_{100}$ in cumulative $N_{800}$ is 89 % and 88 % for sites N322 and S229.

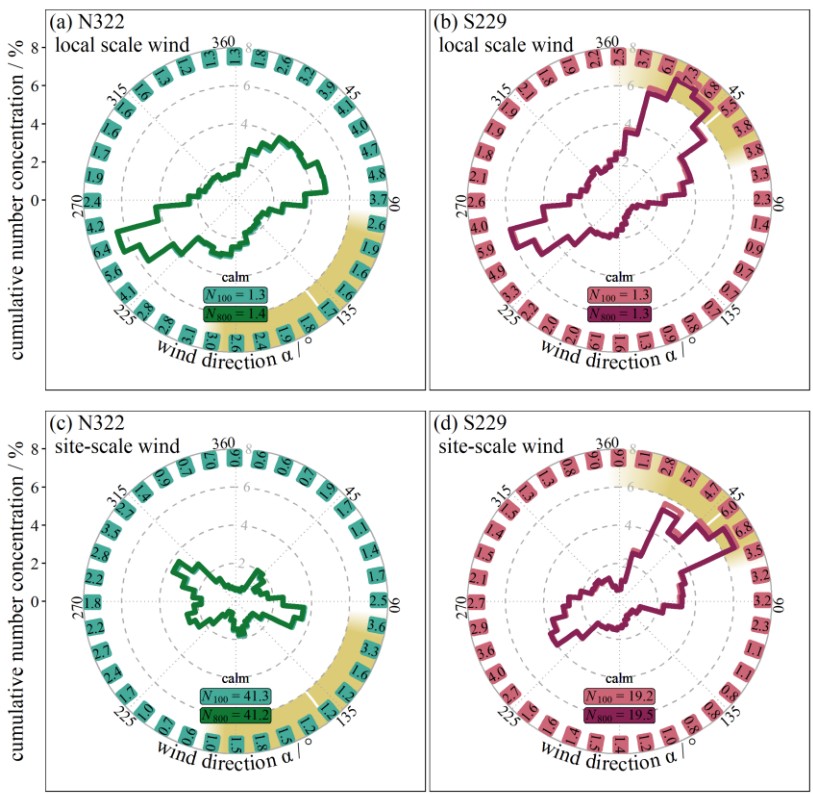

Figure 12: Comparing cumulative concentration roses based on local scale wind, source DWD, (top row) and site-scale wind (bottom row) cumulative concentration roses for sites N322 in Freising (a and c) and S229 in Hallbergmoos (b and d), during the entire observation period for (a) and (b), and for November 2021 to July 2022 for (c) and (d), see Fig. 5. The outer ring contains the values for the $N_{100}$ fraction in each of the 35 bins, calms are noted extra. The revolving labels are centered with the bins. See Table 2. The yellowish arc is the airport sector indicator, see Fig. 1

To quantify the extent that northern and southern adjacent residential areas were potentially impacted by UFP advected from Munich Airport, the normalized cumulative number concentrations again were displayed as a function of wind direction in Fig. 12. This way we can account for both: (1) the frequency of wind directions, as relatively more frequent wind directions

are adding proportionally more to the cumulative particle number concentrations measured at each station, and (2) the elevation of particle number concentrations, as we previously observed with the concentration roses for certain wind directions.

At first glance, the cumulative concentration roses appear more similar to the corresponding wind roses (Fig. 12), than to the concentration roses (Fig. 7 and 9). For the entire year and considering the local wind field based on data from DWD1262, at

site N322 in Freising 21 % of all $N_{100}$ particles were sampled when wind was from Munich Airport (Table 2, Fig. 12, top row).



78 % of all $N_{100}$ particles were sampled when wind came from other wind directions. The cumulative concentration rose shapes differently, when considering the wind data directly measured at each site (Fig. 12, bottom row). In this analysis, 18 % of all $N_{100}$ particles were collected during times with wind directions including Munich Airport, while 41 % could be attributed to other wind directions and 41 % to calm conditions.


For site S229 in Hallbergmoos, local and site-scale wind data overlap better which can likely be explained by the more open surroundings of the site with similar conditions as the DWD station at Munich Airport. Here, the fractions of $N_{100}$ particles sampled with wind from the airport sector and with wind from all other directions were 40 % and 59 % when considering local wind data from DWD1262. They were 31 % and 50 % when considering wind data measured at the site, with 19 % of all $N_{100}$

particles collected during calm conditions.

We consider these values from the sector including Munich Airport as *upper limit* of the airport's possible contribution to the overall UFP mixture at a site. This is because parts of these particles may be transported along with the air prior to approaching Munich Airport and other parts might have been introduced by additional sources between the airport and our sites. For

interpretation, we can combine both stations data in order to exclude the particles that originate from sources prior to Munich Airport and estimate a background level, a *lower limit*.

For the example of Freising, the northern site N322, the wind directions with potential impact by the airport are 95° to 195°, see Fig. 1. We can use the observed particle number concentrations at the southern station S229 in this particular sector as a

background at site N322 for the following reasons: (1) Both stations recorded almost the same cumulative number of particles during the course of the observation period and therefore are comparable. (2) Our comparison between local and site-scale wind data showed agreement within the variability of the data set when considering the entire observation period of one year. Hence, we assume linear advection within a homogeneous, local wind field as main driver for particle transport. (3) The background of both sites is assumed to be similar, because no significant other particle sources or only sources that contribute

to the background of Freising and Hallbergmoos, exist in close proximity prior to the airport for winds approaching from the southeast or the northeast. The background or *lower limit* of site S229 is calculated vice versa. Differences between *upper* and *lower limit* give an estimate on the possible airport contribution to the annual cumulative particle concentration.

Table 2: Upper and lower limits of contribution for the sector including Munich Airport to cumulative particle number concentrations based
on local scale wind for certain size fractions. See also Fig. 12.

| site | *10 to 30 nm / %* | | | *10 to 100 nm / %* | | | *10 to 800 nm / %* | | |
|---|---|---|---|---|---|---|---|---|---|
| | upper | lower | max. MUC | upper | lower | max. MUC | upper | lower | max. MUC |
| N322 | 24 | 9 | 15 | 21 | 11 | 10 | 20 | 11 | 9 |
| S229 | 44 | 24 | 20 | 40 | 26 | 14 | 39 | 26 | 13 |



This estimate is a maximum value, since the lower limit cannot account for sources that are in between the airport and the measuring sites.

For site N322, we calculate *lower and upper limit* as 11 % and 21 %. Thus, out of all UFP monitored in Freising in August 2021 to July 2022 up to 10 % were likely directly originating from Munich Airport and the surrounding infrastructure. For site S229, *lower* and *upper limit* were calculated as 26 % and 40 %. This means that out of all $N_{100}$ particles detected in Hallbergmoos 26 % approached from the wind directions attributed to the airport but are possibly background. Therefore, the airport likely contributed up to 14 % of all UFP that were measured during the observation period. A summary of *upper* and *lower limits* for possible direct airport contribution is given in Table 2. Here the fraction of particles with smaller mobility diameters (10 to 30 nm) has a relatively higher fraction of cumulative particle concentrations attributable to Munich Airport with 15 % and 20 % for Freising and Hallbergmoos, respectively.

## 4     Conclusions

In this study, we presented the first UFP observations in adjacent residential areas around Munich Airport covering one year from August 2021 to July 2022. Our setup presents a novel perspective on the ongoing discussion about the extent to which airports can impact the local air quality: We conducted simultaneous measurements at two sites N322 (Freising) and S229 (Hallbergmoos) in close proximity to the airport, that were neither covering prevailing wind directions nor experiencing any fly overs. Hence, the sole import of UFP from the airport into the adjacent residential areas was possible through advection from potential emission sources of ground-based activities or take-off and landing. With this novel approach, we explored three important aspects:

1. **Overall potential impact of the airport on air quality in adjacent residential areas**: Both locations had a similar airborne particle load, as integrating over the entire measurement period of one year showed great agreement between the observed particle number concentrations at both sites for mean, median and cumulative values. The annual median $N_{800}$ particle number concentrations of 6150 cm$^{-3}$ (N322, Freising) and 5860 cm$^{-3}$ (S229, Hallbergmoos) was similar to other German urban background stations. Considering local wind, we found evidence of the airports impact on the local air quality (1) from increases of $N_{800}$ and $N_{100}$ particle number concentrations for wind directions facing the airport about a factor of 1.5 to 1.8 for the annual median, (2) from a shift of the modal maximum towards smaller mobility particle diameters associated to wind directions including the direction of Munich Airport, and (3) from a dependence of particle number concentrations and the size distribution on the airports operation hours. We also found that the analysis of the combined dataset of particle number concentrations monitored at the two sites and the local wind data monitored at the airport had limitations. Although the dataset covered one year and therefore provides a statistically sound basis for analysis, a high variability persists and the assumption of a local, homogeneous wind field



likely added inaccuracy to the conclusions. For this reason, we next explored diurnal and seasonal effects, as well as the site-specific wind data.

545

2. **The site-specific spatio-temporal characteristics of the airport's impact on local air quality:** For the entire year, local and site-scale concentration roses exhibited a similar shape and elevation ratio for particle number concentrations towards wind directions facing the airport. However, when we resolved the details of diurnal and seasonal variations a potential impact of the airport on the two sites atmospheric particle concentrations became less evident. The diurnal variation of particle number concentrations was typical for an urban air mixture displaying both local sources and atmospheric boundary layer development. The concentration roses featured a distinct seasonality with a potential effect of vegetation on the surface roughness, hence wind speed and flows. This in turn likely affects the simplified view of an advection driven particle transport. Over the time period of one year, this might be negligible, however in light of many field campaign-based studies covering limited time periods, these site-specific alterations should be taken into account.

3. **Upper and lower limits of potential UFP advection from Munich Airport into the adjacent residential areas:** We introduced here a cumulative concentration rose, which displays the number concentration of UFP that a person would be exposed to when standing at the measuring sites N322 or S229 around Munich Airport for one year. This approach allowed us to not only explore the elevation of particle number concentrations for one wind direction in comparison to another, but also provided a measure for the relative frequency of wind directions as they occurred during the observation period. Up to 21 % of all UFP measured in Freising were measured during wind that was from the airport's direction. In the south, this fraction was higher with about 40 %. However, we found that there was a background of 11 % in Freising and 26 % on Hallbergmoos. Hence, likely the range of UFP advection from Munich Airport to the measuring sites in Freising and Hallbergmoos, was up to 10 % and 14 %. With this novel approach we provided evidence for the contribution of the airport to the air quality of adjacent residential areas. We could account for the stations background to estimate the relative magnitude of this contribution. However, UFP sources situated between airport and measuring sites, such as traffic on the highway and related infrastructure, cannot be separated from this.

570

Note that during the observation period analysed here, air traffic was still below pre-COVID-19 pandemic years in terms of flights. The airport's impact on local air quality will likely increase with increasing numbers of flights. Nevertheless, this study highlighted the importance of exploring the spatio-temporal variability of the combined view on particle number concentrations, size distributions, and wind direction frequency in detail. While further insight into the details of the underlying processes remains to be derived, we assessed here for the first time the potential impact of UFP from Munich Airport on the air quality of adjacent residential areas.



*Author contributions:* AN did the funding acquisition, conceptualization of the project and provided supervision. AN and JS were responsible for project administration and developed the methodology. JS and MF conducted the investigation, operated the measurement stations and assured validation of the results. JS was in charge of data curation, formal analysis, visualization and wrote the scripts and packages utilized for these tasks, with support from CT and AN. JS and AN prepared the manuscript with contributions from all other co-authors.

*Competing interests:* The authors declare that they have no conflict of interest.

*Acknowledgements:* This work is financed by the Bavarian State Ministry of the Environment and Consumer Protection, project TLK01U-76519. We thank the cities of Freising and Hallbergmoos for their support during the site surveying and the provision of the installation areas for our measurement stations. In addition, we thank the city of Freising for the ongoing financial support of the project. We would like to thank Kay Weinhold from the Leibniz Institute for Tropospheric Research and Sebastian Schmitt from TSI Germany GmbH for their support regarding operation and quality assurance of MPSS and CPC devices. Also, we thank Alexander Vogel from the Goethe-University Frankfurt and Martin Obst from the University of Bayreuth for the critical discussion of the results. We thank Mike Pitz and Adam Mühlbauer from the Bavarian Environment Agency for the exchange on and technology of UFP measurement stations. Further, we thank the mechanical workshop at faculty II of the University of Bayreuth for manufacturing all the customised and special parts of our measuring containers. A special thank you goes to Agnes Bednorz and Andrej Einhorn, representing all the support we received from members of the Atmospheric Chemistry group at the University of Bayreuth. Funded by the Deutsche Forschungsgemeinschaft (DFG, German Research Foundation) – 491183248. Funded by the Open Access Publication Fund of the University of Bayreuth.

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
