# Peer review of "Introducing the novel concept of cumulative concentration roses for studying the transport of ultrafine particles from an airport to adjacent residential areas."

_EGUsphere, 2023_

## Author Response (AR1)

**Referee's comments/questions in bold upright.**
Authors' answers in regular upright.
*Citations from and changes to the pre-print in regular italic.*

In the following, the refrees' comments/questions are addressed one by one. For improving the manuscript, we incorporated small changes to the text, which are highlighted accordingly.

We address one comment/question per page.

All line numbers refer to the preprint published at
https://egusphere.copernicus.org/preprints/2023/egusphere-2023-1696/egusphere-2023-1696.pdf

**Referee's comments/questions in bold upright.**
Authors' answers in regular upright.
*Citations from and changes to the pre-print in regular italic.*

Please note, that there is one set of changes that is NOT based on the refrees' comments/questions, but corrects a minor error that was found by the authors when creating this revised version of the manuscript. These changes are immediately handled on the following page 2.

https://egusphere.copernicus.org/preprints/2023/egusphere-2023-1696/

**Referee's comments/questions in bold upright.**
Authors' answers in regular upright.
*Citations from and changes to the pre-print in regular italic.*

In Section 2.4 we introduce the used wind data that we obtained from Germany's National Meteorological Service. In lines 234 to 235 we state the following:

*The DWD reports wind data based on a wind rose with 35 bins ranging from 10 to 360ᶜ. The actual bin width is -5.0°/+4.9ᶜ.*

Here we reported the number of bins incorrectly: a range from 10° to 360° with a bin width of -5.0°/+4.9° (basically a 10° angular resolution) yields 36 bins, not 35 bins. We propose the following change for lines 234 to 235:

Original version:
*The DWD reports wind data based on a wind rose with 35 bins ranging from 10 to 360°.*

New version:
*The DWD reports wind data based on a wind rose with 36 bins ranging from 10 to 360°.*

We propose the following change for line 241:

Original version:
*[…] using the same 35 bins and reporting […]*

New version:
*[…] using the same 36 bins and reporting […]*

We propose the following change for line 507, the caption of Fig. 12 in the preprint, which became Fig. 11 in the revised version:

Original version:
*[…] in each of the 35 bins, calms are noted extra […]*

New version:
*[…] in each of the 36 bins, calms are noted extra […]*

The discovered error and the proposed changes to the manuscript have NO influence on the data processing, the presented results and their interpretation.

https://egusphere.copernicus.org/preprints/2023/egusphere-2023-1696/

**Referee's comments/questions in bold upright.**
Authors' answers in regular upright.
*Citations from and changes to the pre-print in regular italic.*

https://doi.org/10.5194/egusphere-2023-1696-RC1

**line 63: There are also potential ground level impacts from landing aircraft due to lateral dispersion enhanced by wingtip vortices.**

This is an important point that was not addressed yet in the manuscript as the results are discussed in light of advection from the airport and the ground-based activities. Nevertheless, we agree that this point should be added to the introduction for completeness, we acknowledge the potential ground level impacts by proposing the following change, inserting on line 63:

Original version:
*[…] Ungeheuer et al., 2022). As airports are typically […].*

New version:
*[…] Ungeheuer et al., 2022). During take-off and approach in particular, wingtip vortices might enhance the dispersion of UFP from very low altitudes towards the ground (Graham and Raper, 2006; Westerdahl et al., 2008; Hudda et al., 2020). As airports are typically […].*

The change proposed above adds the following two publications to the References section:

*Graham, A. and Raper, D. W.: Transport to ground of emissions in aircraft wakes. Part I: Processes, Atmospheric Environment, 40, 5574–5585, https://doi.org/10.1016/j.atmosenv.2006.05.015, 2006.*

*Hudda, N., Durant, L. W., Fruin, S. A., and Durant, J. L.: Impacts of Aviation Emissions on Near-Airport Residential Air Quality, Environmental Science & Technology, 54, 8580–8588, https://doi.org/10.1021/acs.est.0c01859, 2020.*

Please note: apparently Zotero deactivates tracking changes immediately before insertion of a citation and re-activates it after insertion. Hence the new citations in the text and References section are NOT highlighted in the track changes in Word.

https://egusphere.copernicus.org/preprints/2023/egusphere-2023-1696/

**Referee's comments/questions in bold upright.**
Authors' answers in regular upright.
*Citations from and changes to the pre-print in regular italic.*

https://doi.org/10.5194/egusphere-2023-1696-RC1
**line 81: Yu et al was based on a 4km spatial resolution model that might not account for the finer spatial scale resolution at locations near the airport under the flight path. This is mentioned late in the reference to the Zurich study and also demonstrated initially further downwind by Hudda et al.**

Thank you for this comment. As this point may further highlight and motivate explicitly why to study the closer surroundings of an airport, we propose the following change to the sentence spanning lines 79-81:

Original version:
*Overall, with a regional chemical transport model study it was calculated that aviation contributed to the ultrafine particulate matter to about 7 % in downtown Los Angeles next to other regional sources such as the consumption of natural gas, on-road-traffic, and cooking (Yu et al., 2019).*

New version:
*For example, Yu et al. (2019) calculated that aviation contributed to the ultrafine particulate matter to about 7 % in downtown Los Angeles next to other regional sources such as the consumption of natural gas, on-road-traffic, and cooking. However, the model was too coarse to resolve the finer spatial scale variation at residential areas within close proximity to the airport.*

https://egusphere.copernicus.org/preprints/2023/egusphere-2023-1696/

https://doi.org/10.5194/egusphere-2023-1696-RC1

**line 248: Where is the airport parking? This is also relevant to the conclusion on line 569.**

We assume that referee #2 refers to cases at other airports where airport parking is concentrated into a single huge parking area or car park. At Munich Airport the airport parking spreads across multiple parking areas and car parks on the premises of the airport and is hence covered by each site's respective airport sector. See the following schematic of the parking areas and parks that was obtained from https://www.munich-airport.com/parking-260363 on 20.09.2023.

[Figure]

To reflect this better in the manuscript we propose the following change:

Attach to line 153: *Parking at Munich Airport spreads across multiple parking areas and car parks on the premises of the airport and is not incorporated to a single large scale car park.*

Regarding the conclusion on line 569: The airport parking is part of the airport as an emission source. What line 569 covers is, that with the described approach we can estimate a site specific airport contribution, but this contribution inevitably includes any source located between airport and site. For site N322 this would be the highway A92 and its feeder in particular.

Referee's comments/questions in bold upright.
Authors' answers in regular upright.
Citations from and changes to the pre-print in regular italic.

https://doi.org/10.5194/egusphere-2023-1696-RC1

**line 299: This paragraph could do a better job of simply describing the cumulative method used here, especially starting on line 306. It took me a while to figure out what was going on.**

This is a very valuable feedback, because the method for deriving cumulative concentration roses is critical for this manuscript. Hence, we hope to improve quick access to the method by modifying the whole paragraph "cumulative number concentrations":

Change on line 299 to 309: *Cumulative concentration rose: The presented data covers the annual cycle of the observation period from August 2021 to July 2022 almost completely with a data availability of 92 % for site N322 and 94 % for site S229. Additionally, no extreme particle emissions were observed and the dataset is a good representation of typical airborne particle loads at the two sites during the course of one year. Hence, we first summed up all particle number concentrations to a cumulative number concentration for $N_{100}$ or $N_{800}$ at each site. The final cumulated particle number concentration represents the total particle load at each site after one year (100 %). Next, the observed particle number concentrations were grouped by the wind direction that occurred at the given time (36 bins + 1 calm bin, see Sect. 2.4). For each bin a cumulative number concentration is calculated and divided by the total particle load. This way the contribution of each bin relative to the total particle load for $N_{100}$ or $N_{800}$ over the course of the observation period can be expressed as fraction in percent.*

In addition, we think that Fig. 11 is not necessary anymore and that we can free additional space by removing it. Hence, we rewrite line 466 by removing the reference to Fig. 11 and instead referring to the methodology section and the cumulative number concentrations as follows:

Original version:
*As the presented data covers the annual cycle of the observation period August 2021 to July 2022 almost completely, it can be used to sum up the observed particle number concentrations to a cumulative value, see Fig. 11.*

New version:
*As the presented data covers the annual cycle of the observation period August 2021 to July 2022 almost completely, it can be used to sum up the observed particle number concentrations to a cumulative value, see Sect. 2.5 on cumulative number concentrations.*

Please note: the pictures are anchored in the word document and not placed "in line with text", which would result in an unnecessary messy and high maintenance document. Due to this, the track changes feature does NOT highlight the deleted Figure 11, only the caption is marked as deleted. In the clean version with accepted changes, Figure 11 is correctly removed.

https://egusphere.copernicus.org/preprints/2023/egusphere-2023-1696/

**Referee's comments/questions in bold upright.**
Authors' answers in regular upright.
*Citations from and changes to the pre-print in regular italic.*

https://doi.org/10.5194/egusphere-2023-1696-RC1
**line 330: need a comma rather than a period.**

For grammatical correctness, a comma instead of a period would be necessary. However, this would result in a sentence spanning three lines. Instead, we propose to modify the first sentence on line 329 as follows. The period then can remain unchanged:

Original version:
*As the number of flights per month [...].*

New version:
*The number of flights per month [...].*

https://doi.org/10.5194/egusphere-2023-1696-RC1

https://egusphere.copernicus.org/preprints/2023/egusphere-2023-1696/

**Referee's comments/questions in bold upright.**
Authors' answers in regular upright.
*Citations from and changes to the pre-print in regular italic.*

<https://doi.org/10.5194/egusphere-2023-1696-RC1>

**line 458: "strong evidence" is a bit too strong. When the wind is coming from the airport, the evidence is clear. It is just that it is not always coming from the airport.**

We now reworded the two sentences, with the aim to discuss our analysis more precisely. We hence propose the following change to lines 458-460 as follows:

Original version:
*Overall, the diurnal variation of UFP number concentrations at the two sites does not provide any strong evidence for the impact of the airport on the local air quality. In contrast, the concentration roses for winds blowing from the direction of the airport had the highest daytime UFP number concentrations.*

New version:
*Overall, it is difficult to assess the impact of the airport on the local air quality from this view on the diurnal variation of UFP number concentrations at the two sites only. Yet, the combined results show that the concentration roses for winds blowing from the direction of the airport had the highest daytime UFP number concentrations.*

<https://egusphere.copernicus.org/preprints/2023/egusphere-2023-1696/>

**Referee's comments/questions in bold upright.**
Authors' answers in regular upright.
*Citations from and changes to the pre-print in regular italic.*

https://doi.org/10.5194/egusphere-2023-1696-RC1
**line 550: at site s229 the effect is evident, although not at the other site.**

We understand that the wording should be more precise and propose the following change to lines 548–550:

Original version:
*However, when we resolved the details of diurnal and seasonal variations a potential impact of the airport on the two sites atmospheric particle concentrations became less evident.*

New version:
*However, when we resolved the details of diurnal and seasonal variations a potential impact of the airport on the two sites became less evident, as it was likely covered by the variability of other sources and atmospheric conditions.*

https://egusphere.copernicus.org/preprints/2023/egusphere-2023-1696/

https://doi.org/10.5194/egusphere-2023-1696-RC2

**lines 174-185 - More detail about the site locations would be helpful. The locations are somewhat compromised by the proximity of local sources. For example, N322 appears to be screened by trees to the south and in the middle of a working gardeners yard. S229 appears to be similarly sheltered by trees and a factory building. As the authors note later in the manuscript, both locations are also influenced by contributions not from the airport.**

We agree with the referee that the possible influences in terms of advection and local non-airport contributions are not fully reflected in a concise manner yet. Hence, we propose the following changes, including additional figures Fig. 6a-d for the supplement (the measurement containers at each site can actually be "seen" in the digital orthophotographs):

Attaching to line 180:
*In order to allow free advection from all wind directions, the locations for both sites were chosen as open as possible. Still due to their positioning within the city areas, both sites are potentially influenced by their closer surroundings.*

For site N322 changing the paragraph from lines 182-185:

Original version:
*Site N322 to the north is located within the southern urban area of Freising on the premises of a city gardener and is 444 m above mean sea level. The nearest surrounding is characterized by a business park and a highway (A92 and its feeder FS45) spanning from south to south-east. A small meadow is located about 50 m to the west. To the north residential areas predominate.*

New version:
*Site N322 to the north is located within the southern urban area of Freising on the premises of a city gardener and is 444 m above mean sea level. In close proximity only low frequency traffic from the gardeners activities are expected. The surrounding is characterized by non-manufacturing business parks in about 60 m distance to the northeast and about 200 m distance to the south and southeast, a highway in about 490 m distance, and its feeder in about 50 m distance with frequent traffic spanning from south to southeast. In that direction, with about 35 m distance a single row of trees and shrubs (max. 6 m in height) borders the premises of the city gardener. A small meadow with tree heights of about 20 m is located about 50 m to the west. From about 250 m to the north and northeast residential areas begin to predominate, see Fig. SI6a and SI6c.*

For site S229 changing the paragraph from lines 187–190:

Original version:
*Site S229 to the south is placed on a wide-open space westerly of the city Hallbergmoos and is 456 m above mean sea level. To the east and northeast residential areas can be found. When looking into airport direction north, a business park is situated between the southern runway and site S229. The surroundings in westerly directions are mostly under agricultural use, but crossed by a federal highway B301 in north-south direction.*

New version:
*Site S229 to the south is placed on a wide-open space westerly of the city Hallbergmoos and is 456 m above mean sea level. To the east and northeast residential areas are located in about 300 m distance. When looking into airport direction north, a single row of young trees and shrubs spans from 310° to 110°. The shortest distance to S229 is about 12 m. The tree heights vary depending on season about 6 m. In the same direction, a non-manufacturing business park is situated between the southern runway and site S229 in about 250 m distance, with the exception of an automobile manufacturing plant. The surroundings in westerly directions are mostly under agricultural use, but are crossed by a federal*

**Referee's comments/questions in bold upright.**
Authors' answers in regular upright.
*Citations from and changes to the pre-print in regular italic.*

*highway B301 in north-south direction. Overall, site S229 can be considered under less road traffic influence than N322, see Fig. SI6b and SI6d.*

[Figure]

*Fig. SI6a: Map section of the immediate surroundings of site N322. Intended to be used together with the information given in Sect 2.2.*

[Figure]

*Fig. SI6b: Map section of the immediate surroundings of site S229. Intended to be used together with the information given in Sect 2.2.*

https://egusphere.copernicus.org/preprints/2023/egusphere-2023-1696/

[Figure]

*Fig. SI6c: Digital ortophotograph (DOP) of the immediate surroundings of site N322. DOP with 0.4 m resolution. Intended to be used together with the information given in Sect 2.2. CC BY 4.0, Bayerische Vermessungsverwaltung – www.geodaten.bayern.de. Labels and scale added by the authors.*

[Figure]

*Fig. SI6d: Digital ortophotograph (DOP) of the immediate surroundings of site S229. DOP with 0.4 m resolution. Intended to be used together with the information given in Sect 2.2. CC BY 4.0, Bayerische Vermessungsverwaltung – www.geodaten.bayern.de. Labels and scale added by the authors.*

https://egusphere.copernicus.org/preprints/2023/egusphere-2023-1696/

**Referee's comments/questions in bold upright.**
Authors' answers in regular upright.
*Citations from and changes to the pre-print in regular italic.*

https://doi.org/10.5194/egusphere-2023-1696-RC2

**lines 375-385 - Was any attempt made to correlate departures/arrivals with measurements? Did the airport provide basic aircraft movements and operation modes for the survey periods? This would help to attribute changes in particle diameters to aircraft activity.**

The referee raises a valid point. Indeed, we do have information about departures and arrivals as well as on the biggest streets traffic. However, we decided to dedicate a second publication to the detailed analysis of these aspects. We find that the focus on the wind-driven transport of UFP in the surroundings of the airport viewed from the observations at our two sites is for itself an interesting and novel aspect. Starting a new point of discussion with further data would have enlarged the content severely and would have moved the scope of this publication. We hope that the reviewer and editor will understand and kindly refer to the fact that the current manuscript already spans over 600 lines of text and 12 figures.

https://egusphere.copernicus.org/preprints/2023/egusphere-2023-1696/

**Referee's comments/questions in bold upright.**
Authors' answers in regular upright.
*Citations from and changes to the pre-print in regular italic.*

https://doi.org/10.5194/egusphere-2023-1696-RC2

**line 400 - Given the possible screening of the sites by vegetation, do the authors think that meteorology measurements at 5,3m will be sufficiently representative of the sampling inlets at ca.3,0m? Will winter and summer leaf growth cause any differences?**

The sampling inlet for UFP is actually 4.2 m above ground level (line 206). The difference in sampling height for UFP data and site-scale meteorological data is hence about 1.1 m. The two sites N322 and S229 were chosen to avoid possible screening through tall vegetation and buildings as good as possible. Nevertheless, as we are located in an urban environment and aimed for measuring at a representative height according to monitoring stations standards, there are some obstacles. The closest objects are:

- meadow of 20 m height 50 m west of N322
- young trees and shrubs of about 6 m height 35 m south of N322
- young trees and shrubs of about 6 m height 12 m north of S229

It is likely that there are seasonal differences due to the greening of the trees. This is suggested from Fig. SI3-5 and discussed briefly in the text in lines 551-553. However, no other tall objects were possibly shielding our stations in any directions.

In the context of this study, we use the 5.3 m wind data to demonstrate the differences in wind direction and speed when moving from the official 10 m measuring height to measuring heights that are closer to the ground and located at the sampling site within the urban street increment. For this purpose, we consider the 5.3 m meteorology measurement to be more representative for our observations than the 10 m airport meteorology. We hypothesized that these differences are important and impact the interpretation of airport studies since they often use the local-scale wind data, which is typically measured at the airport in 10 m height. However, in lines 545-555 *(comparing Fig. 7 and 9) we conclude that for the entire year, the overall essence of information on the UFP transport derived from the local and site-scale wind data is mostly comparable. Yet, when analysing smaller-scale temporal and spatial variability (see Fig. SI3-5), the site-scale wind data reveals more insight. For example, Fig. 12 shows a better overlap of local and site-scale cumulated concentration roses for the southern station S229. This is likely due to the open position of the station particularly regarding the main wind directions. Contrastingly, we can observe at N322 that the small meadow in about 50m west to the station seems to act as sufficient screen for wind and hence related particle transport. This is also visible in Figure 2 (h) and (k) where the site scale wind roses for spring and summer at N322 are presented.*

https://egusphere.copernicus.org/preprints/2023/egusphere-2023-1696/

**Referee's comments/questions in bold upright.**
Authors' answers in regular upright.
*Citations from and changes to the pre-print in regular italic.*

https://doi.org/10.5194/egusphere-2023-1696-RC2

**line 480-484 - Is it possible to disaggregate the contributions of road traffic from the airport related wind direction for N322? Diurnal traffic flow and composition information might help**

Please, see our answer regarding the comment on lines 375-385.

https://egusphere.copernicus.org/preprints/2023/egusphere-2023-1696/

**Referee's comments/questions in bold upright.**
Authors' answers in regular upright.
*Citations from and changes to the pre-print in regular italic.*

https://doi.org/10.5194/egusphere-2023-1696-RC2

**Figures 6 - 8 suggest that arriving and departing aircraft, once they are airborne and beyond the airfield boundaries, have very little influence on measured ground level UFP. Is it worth mentioning this? It's an important observation that is contrary to many other studies.**

Yes, interestingly this seems to be true for most of our observations. The analysis of the median modal particle diameter in Figure 8 shows that the southern station S229 exhibits relatively small particles, mostly smaller than 20 nm, for northernly and westerly wind directions. These include the airport, but also possibly airplanes during take-off to southern destinations. Please, find here the respective description in the manuscript:

*Lines 379-381: Particularly for the southern site in Hallbergmoos a wide range of wind directions was associated with such small modal diameters. Possibly, this is related to airplanes taking-off during west winds to southern destinations (see Fig. 1 and corridor 26R/L-S/SO).*

https://egusphere.copernicus.org/preprints/2023/egusphere-2023-1696/